# A first-takes-all model of centriole copy number control based on cartwheel elongation

**Marco António Dias Louro**[1]*, **Mónica Bettencourt-Dias**[1], **Jorge Carneiro**[1,2]*

**1** Instituto Gulbenkian de Ciência, Oeiras, Portugal, **2** Instituto de Tecnologia Química e Biológica António Xavier, Universidade Nova, Oeiras, Portugal

* mlouro@igc.gulbenkian.pt (MADL); jcarneir@igc.gulbenkian.pt (JC)

**Data Availability Statement:** All relevant data are within the manuscript and its Supporting information files.

## Abstract

How cells control the numbers of subcellular components is a fundamental question in biology. Given that biosynthetic processes are fundamentally stochastic it is utterly puzzling that some structures display no copy number variation within a cell population. Centriole biogenesis, with each centriole being duplicated once and only once per cell cycle, stands out due to its remarkable fidelity. This is a highly controlled process, which depends on low-abundance rate-limiting factors. How can exactly one centriole copy be produced given the variation in the concentration of these key factors? Hitherto, tentative explanations of this control evoked lateral inhibition- or phase separation-like mechanisms emerging from the dynamics of these rate-limiting factors but how strict centriole number is regulated remains unsolved. Here, a novel solution to centriole copy number control is proposed based on the assembly of a centriolar scaffold, the cartwheel. We assume that cartwheel building blocks accumulate around the mother centriole at supercritical concentrations, sufficient to assemble one or more cartwheels. Our key postulate is that once the first cartwheel is formed it continues to elongate by stacking the intermediate building blocks that would otherwise form supernumerary cartwheels. Using stochastic models and simulations, we show that this mechanism may ensure formation of one and only one cartwheel robustly over a wide range of parameter values. By comparison to alternative models, we conclude that the distinctive signatures of this novel mechanism are an increasing assembly time with cartwheel numbers and the translation of stochasticity in building block concentrations into variation in cartwheel numbers or length.

## Author summary

Centriole duplication stands out as a biosynthetic process of exquisite fidelity in the noisy world of the cell. Each centriole duplicates exactly once per cell cycle, such that the number of centrioles per cell shows no variance across cells, in contrast with most cellular components that show broadly distributed copy numbers. We propose a new solution to the number control problem. We show that elongation of the first cartwheel, a core

**Funding:** MADL is funded by the PhD Programme in Integrative Biology and Biomedicine of the Instituto Gulbenkian de Ciência and ITQB NOVA and also by the Fundação para a Ciência e a Tecnologia (ref PD/BD/139217/2018). JC and MBD acknowledge the funding of Fundação Calouste Gulbenkian and Fundação para a Ciência e a Tecnologia (ref. UID/Multi/04555/2013). MBD was funded by the European Research Council (ERC) consolidator grant 683258-CentrioleBirthDeat. The funders had no role in study design, data collection and analysis, decision to publish, or preparation of the manuscript.

**Competing interests:** The authors have declared that no competing interests exist.

centriolar structure, may sequester the building blocks necessary to assemble supernumerary centrioles. As a corollary, the variation in regulatory kinases and cartwheel components across the cell population is predicted to translate into cartwheel length variation instead of copy number variation, which is an intriguing overlooked possibility.

## Introduction

Cellular biosynthetic processes are inherently stochastic due to the low abundance of the key intervening molecules. Importantly, variation in the concentrations of specific proteins is systematically observed over time in single cells [1] and within populations of genetically identical cells in the same state [2]. In this context, it is paradoxical that some cellular components have little or zero variance in copy number across a cell population. This suggests an underlying biosynthetic process of exquisite fidelity, whose deterministic-like control mechanism demands explanation.

Centriole duplication stands out as a paradigm of an extraordinary number control mechanism [3, 4]. Typical proliferating vertebrate cells contain two centrioles in G1. During S-phase, these centrioles are duplicated once and only once. After mitosis, each daughter cell inherits a single pair of centrioles. Centriole over- or underproduction generally leads to mitotic defects and cell-cycle arrest or cell death. [5–7]; thus, rarely observed in healthy tissues.

In human cells, centriole numbers depend on the levels of key proteins, namely Plk4, STIL, and SAS-6. Depletion of either one efficiently abrogates centriole duplication and their overexpression triggers the production of random supernumerary centrioles in a dose-dependent manner [8–11]. Quantitative proteomics revealed that Plk4, STIL, and SAS-6 are relatively low-abundance proteins at physiological conditions [12] although centriole duplication proceeds efficiently at a relatively broad range of their concentrations [13, 14]. Although active degradation of Plk4, STIL, and SAS-6 via the cell cycle machinery and/or their own activity [9, 10, 15–18] may allow their levels to be kept within a given range, quasi-deterministic centriole duplication requires mechanisms that buffer stochasticity in Plk4, STIL, and SAS-6 levels.

Several studies proposed that Plk4 and STIL have an essential role in controlling centriole copy numbers by accumulating the essential components in a local focus around the mother centriole [19–23]. Mathematical models describing these focusing dynamics based on different mechanisms have been put forward [14, 23–25]. These models were formulated in terms of the local Plk4, STIL, and SAS-6 concentrations, which are continuous quantities untranslatable to discrete centriole numbers. To overcome this basic limitation [14] and [24] defined discrete spatial compartments around the mother centriole, each being able to harbour one and only one centriole by construction. This limitation notwithstanding, the focused accumulation explains naturally the prevention of ectopic centrioles by the maintenance of the key components concentrations below critical values elsewhere in the cell and around the mother centriole. Supercritical concentrations of these components are necessary to ensure that at least one centriole is formed in either normal or ectopic foci, but are also sufficient to enable formation of extra centrioles (as readily observed in overexpression experiments [26]). Thus, these (qualitative or quantitative) proposals fail to explain how the formation of supernumerary centrioles is avoided at the foci, where Plk4, STIL and the other components accumulate at supercritical concentrations. This is the control problem that is solved by the first-takes-all model proposed here.

We posit that centriole number control can be achieved by controlling the numbers of a core structure called the cartwheel. The cartwheel is assembled following Plk4, STIL, and

SAS-6 focusing [18, 19, 22] and acts as a platform around which one and only one centriole is built. The central hub of the cartwheel consists of a multi-layered stack, where each ring-like layer is composed of nine SAS-6 homodimers [27–32]. Ring formation via oligomerisation and stacking depend on structural properties of SAS-6, at least partly [30, 33]. We hypothesise that growing SAS-6 oligomers can be stacked up on the first-formed cartwheel hub, heretofore simply referred to as cartwheel, thus preventing the formation of supernumerary structures.

We assess this hypothesis by modelling cartwheel assembly and elongation as a stochastic process. Our model can be summarised as the competition between two processes: the combinatorial multi-step oligomerisation leading to new cartwheels, and the sequestering of intermediate oligomers by previously assembled cartwheels, resulting in new rings being added to the stack. We show that elongation of a single cartwheel, by stacking, can efficiently prevent supernumerary cartwheels over a wide range of model parameters and in distinct theoretical scenarios. Finally, we compare our model to a general mechanism of structure number control by negative feedback and find a set of specific predictions: first, cartwheel assembly time is increasingly delayed with the formation of each extra cartwheel; and second, both cartwheel numbers and length may vary as a function of SAS-6 levels.

## Results

### Sequestering of SAS-6 intermediates inhibits formation of supernumerary cartwheels

We explored a simple principle of biosynthetic regulation, where the the final product of a multi-step assembly pathway inhibits the synthesis of further products by incorporating intermediate molecules as they form. As per our hypothesis, this boils down to a competition between assembly of new cartwheels by oligomerisation of SAS-6 homodimers, and elongation of the first-formed cartwheel by stacking up of these oligomers, in a way that leads to the consecutive addition of rings to the stack (Fig 1).

We first illustrate this principle by calculating the fate of a SAS-6 dimer that may either integrate the first-formed cartwheel or give rise to an extra one. We assume that SAS-6 oligomers $C_i$, containing $1 \leq i < r$ dimers, can grow by irreversible incorporation of a free SAS-6 homodimer, which occurs at a constant rate $k_{on}$—the "oligomerisation" rate, yielding a $C_{i+1}$ oligomer. Upon reaching $C_r$, we consider that an independent cartwheel ring has formed. The completion of this new cartwheel ring is the minimal condition for initiating the synthesis of an individual centriole. Alternatively, at each of the $C_i$ intermediate steps, the oligomers can be sequestered by binding to the previously formed cartwheel also with a constant rate $k_s$—the "stacking" rate.

Consider one SAS-6 homodimer ($C_1$). Given enough time it will either give rise to a new independent cartwheel or be sequestered by the one that is already assembled. As time approaches infinity, the probability that the dimer is incorporated into a new cartwheel ring $C_r$ tends to a maximum:

$$\lim_{t \to \infty} Pr(C_r = 1, t) = \left( \frac{k_{on}}{k_{on} + k_s} \right)^{r-1}, \tag{1}$$

since the dimer will have either formed a new cartwheel or stacked on top of the pre-existing one. The derivation of Eq 1 is provided in Models and methods. The right-hand side of the equation represents the probability that the original dimer undergoes $r - 1$ consecutive oligomerisation steps without any stacking reaction event, thus yielding a new complete cartwheel

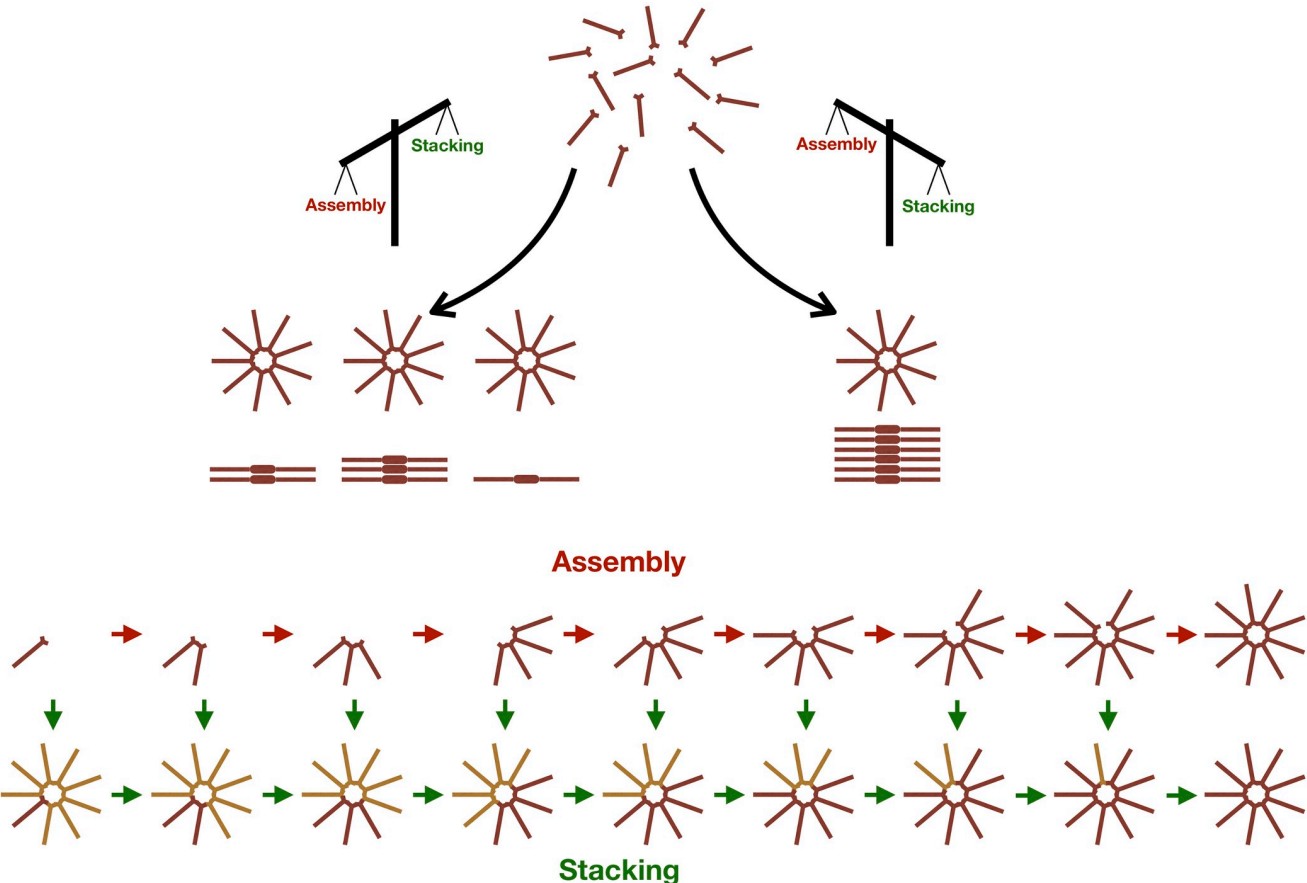

**Fig 1. General scheme of the model.** The SAS-6 dimer is the basic subunit of the carwheel. Higher-order oligomers can arise through oligomerisation of SAS-6 dimers. Once oligomers grow to assemble a complete ring, it is assumed that an independent cartwheel is formed. After the first cartwheel is formed it can elongate by stacking intermediate oligomers onto the top ring layer. Thus, stacking can deplete the pool of available dimers and oligomers necessary to assemble supernumerary cartwheels.

ring. The value of the probability given by Eq 1 decreases exponentially with ring size $r$ and as a power-law function of the stacking rate $k_s$ (Fig 2). In nature, cartwheels display a highly conserved nine-fold symmetry ($r = 9$), although rings of different sizes have been observed *in vitro* [34, 35]. In these conditions, when $k_{on} = k_s$, i.e. when oligomerisation and stacking rates are identical, the probability of forming supernumerary cartwheels is approximately 0.4%. Interestingly, even if the value of $k_s$ is half that of $k_{on}$, this values rises only to approximately 3.9%. Thus, under this simple model, efficient cartwheel number control does not require relatively higher stacking rates. We emphasise that this is possible due to the number of intermediate steps before a new cartwheel ring is assembled.

Despite its quantitative insights, the previous analysis is an oversimplification of the biochemical processes involved in cartwheel assembly. Importantly, ring formation has been described as a combinatorial oligomerisation process, where intermediates of different sizes can combine into higher-order structures [33, 35, 36]. For example, two molecules containing four ($C_4$) and five dimers ($C_5$) can combine, skipping over multiple steps in the assembly pathway, to form a complete ring. This also implies considering an entire molecular population,

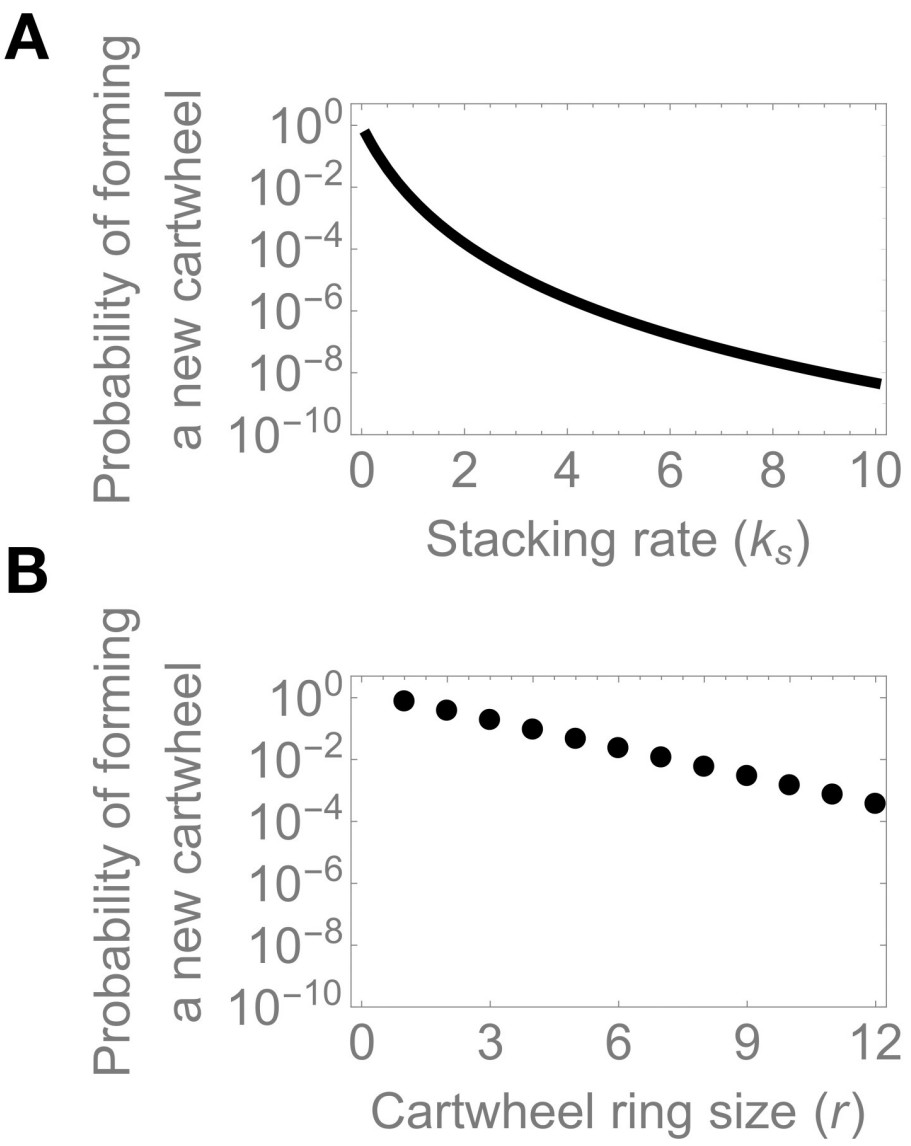

**Fig 2. The probability of forming extra cartwheels depends negatively on the stacking rate and ring size.** Eq 1 was evaluated numerically at the indicated parameter values. We set $k_{on} = 1$.

rather than tracking the fate of a single molecule. To explore these potential issues, we formulated a more realistic model describing the reaction system of cartwheel assembly.

We consider that the reaction system is contained in the focal volume around a pre-existing mother centriole. Fluorescence measurements of SAS-6 intensities at centrioles indicate that it accumulates during S-phase, peaking in G2 [12, 37]. As mentioned above, cytosolic SAS-6 has been found preferentially in homodimer form [37]. In accordance with these observations we assume that SAS-6 expression, degradation, dimerisation, and recruitment to the centrosome through the action of Plk4 and STIL, result in a constant influx of SAS-6 homodimers, $C_1$, into the focal volume, at a rate $\sigma$. $C_i$ and $C_j$ oligomers can interact to produce higher-order $C_{i+j \leq r}$ structures, at a constant rate $k_{on}$. As before, oligomers containing less than $r = 9$ SAS-6 homodimers represent intermediates in the process of cartwheel formation, and oligomers with $r$

dimers represent complete cartwheel rings. In turn, $C_{i+j<r}$ can dissociate into $C_i$ and $C_j$ at a constant rate $k_{off}$.

Let $C_i^*$ represent stacks containing at least one layer of complete rings containing $r = 9$ dimers, topped by an incomplete ring containing $0 \leq i < r$ oligomers. Note that $C_0^*$ substitutes $C_r$. We postulate the existence of a stacking mechanism analogous to ring assembly by oligomerisation. When a $C_i^*$ stack binds a freely-diffusing $C_j$ oligomer it forms a $C_{i+j}^r \leq r$ stack. This can be visualised as a ring being assembled on top of a stack, aligned with the layers underneath [31]. We assume that all oligomerisation reactions yielding complete cartwheel rings as well as all stacking reactions are irreversible.

This model depicts cartwheel assembly as an oligomerisation and stacking process, under a constant influx of SAS-6 dimers. Unlike Eq 1, it describes explicitly the assembly of multiple cartwheels and their elongation. Since there is a constant influx of SAS-6 homodimers into the focal volume, the total number of molecules in the focal volume increases monotonically as a function of time, i.e. it never decreases. Thus, the model does not reach a steady-state, making it quite cumbersome for deriving analytical solutions. Readers may refer to Models and methods for detailed formulation of the model.

We resorted to numerical simulations to study the behaviour of this model. We set a fixed time window in which a cell is permissive to centriole biogenesis, roughly corresponding to S-phase, in which SAS-6 is shuttled into the centrosome. Thus, neither $C_i$ intermediates nor cartwheels are present at the beginning of the simulations. In addition, this poses a challenge to our hypothesis as stacking may not be efficient enough to prevent other large oligomers (e.g. $C_i$) from seeding extra cartwheels. The independent simulations run for each parameter set represent the time courses of the reactions in an independent focal volume around pre-existing mother centriole. Due to the stochastic kinetics, the number and length of $C_i^*$ stacks at the end of the simulations are variable corresponding to the numbers and lengths of the cartwheels across a cell population.

Fig 3 shows the temporal dynamics of cartwheel formation and numbers of $C_i$ intermediates. In the absence of stacking (Fig 3A and 3D), supernumerary cartwheels rapidly appear and keep forming, on average, at an approximately linear rate. When all four parameters are set to the same value (Fig 3B and 3E), the emergence of the first cartwheel coincides with the depletion of $C_i$ intermediates, and supernumerary cartwheels are formed only in 2.3% of the simulations. When $\sigma$ is increased with respect to the other parameters (Fig 3C and 3F), mimicking, for example, SAS-6 overexpression, supernumerary cartwheels appear at a decelerating rate, while reducing the levels of building blocks. See also S1 Fig. Concomitantly, assembled cartwheels elongate continuously (Fig 4 and S2 Fig). For reference, note that in the absence of stacking, cartwheels attain a maximum length of one ring. When stacking is enabled, building blocks are equally distributed by existing cartwheels. This can be observed in "overexpression" conditions (Fig 4C). At first, when the first cartwheel is present, its elongation rate is maximal. When the second cartwheel is formed in all simulations, and formation of additional cartwheels becomes less favourable, the average elongation rate for both cartwheels equilibrates. The difference in assembly times dictates that the first cartwheel is longer, on average, than the second cartwheel, at the end of the simulations. Notwithstanding, inherent stochasticity in molecular influx and stacking result in a time-dependent increase in cartwheel length variability.

In qualitative terms, we obtain similar results as in the previous model. First, stacking can efficiently suppress the formation of supernumerary cartwheels without the need of high stacking rates. Second, we observe a similar dependence on ring size on the probability of forming supernumerary cartwheels (S3 Fig).

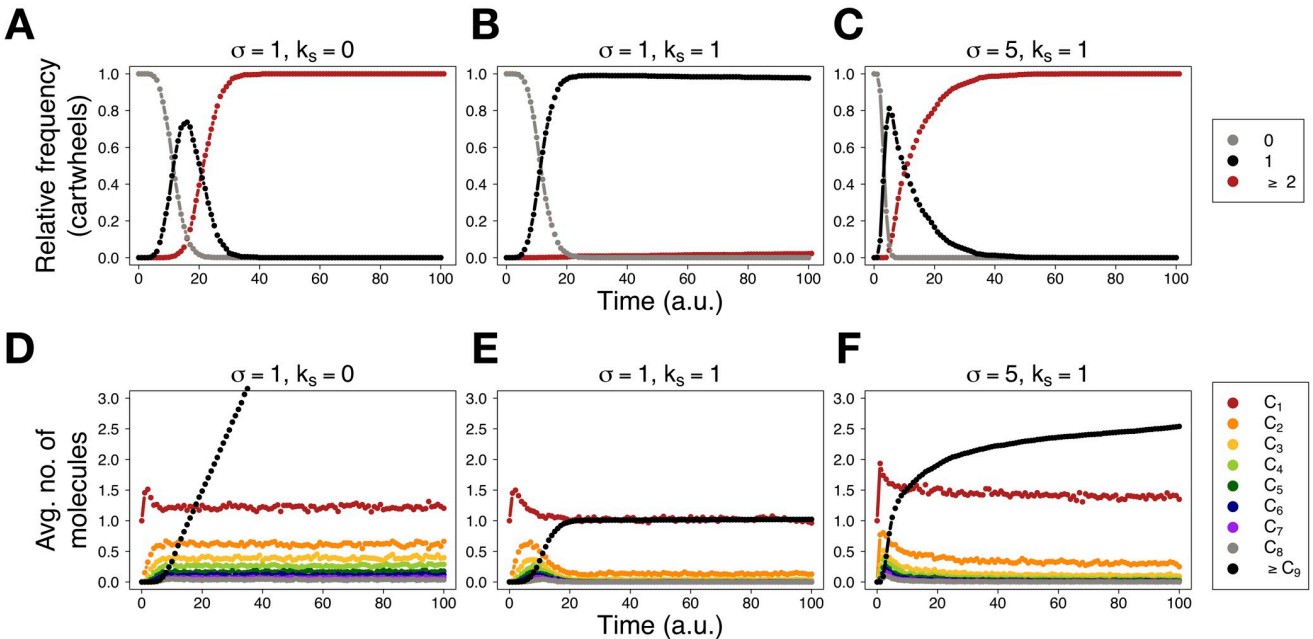

**Fig 3. Stacking causes depletion of intermediates and inhibits the formation of supernumerary cartwheels.** (A-C)—relative frequency of simulations containing 0 (grey), 1 (black), and 2 or more (red) cartwheels as a function of time, for the indicated values of $\sigma$ and $k_s$. (D-F)—average number of molecules across all simulations, as a function of time, for the indicated values of $\sigma$ and $k_s$. We performed 1000 simulations with no SAS-6 molecules initially present and stopped them at time $t = 100$; time $t$ is measured in arbitrary units (a.u.). For all simulations, $k_{on} = k_{off} = 1$. Note that $k_s = 0$ represents absence of stacking. See also Models and methods section for default initial and stopping conditions, and parameter values for the simulations.

## Formation of one and only one cartwheel is achieved with high fidelity for a wide range of parameter values

Up to this point, we have established a simple principle of inhibition-by-stacking. The postulated mechanism can ensure formation of one and only one cartwheel with high fidelity in unbiased conditions in which stacking is not kinetically favoured relatively to the other

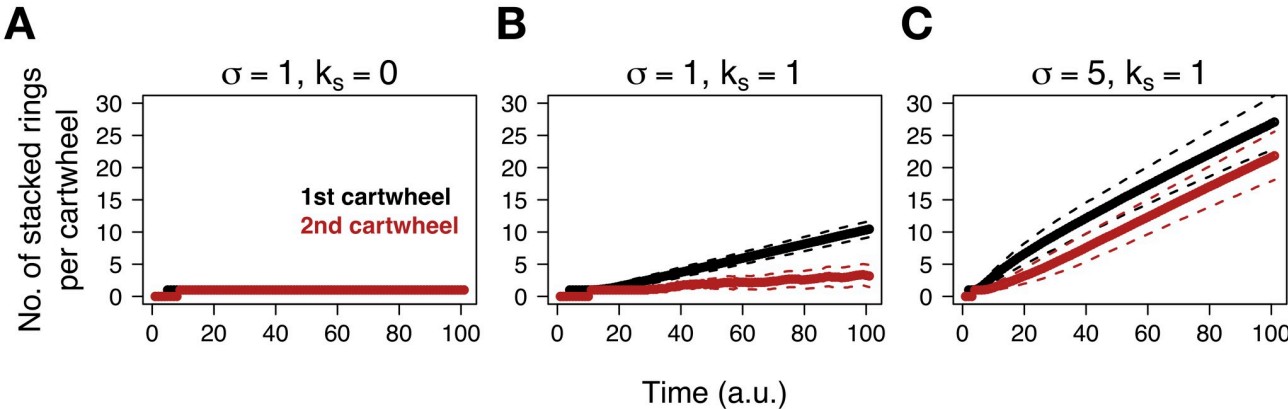

**Fig 4. Cartwheels elongate continuously and show a distribution of lengths along time.** The average number of complete rings present in the first (black) and second (red) cartwheels across all simulations are plotted as a function of time, for the indicated values of $\sigma$ and $k_s$. The dashed lines indicate the average ± standard deviation. We used default simulations conditions as indicated in Fig 3 and in the Models and methods section. Note that $k_s = 0$ represents absence of stacking.

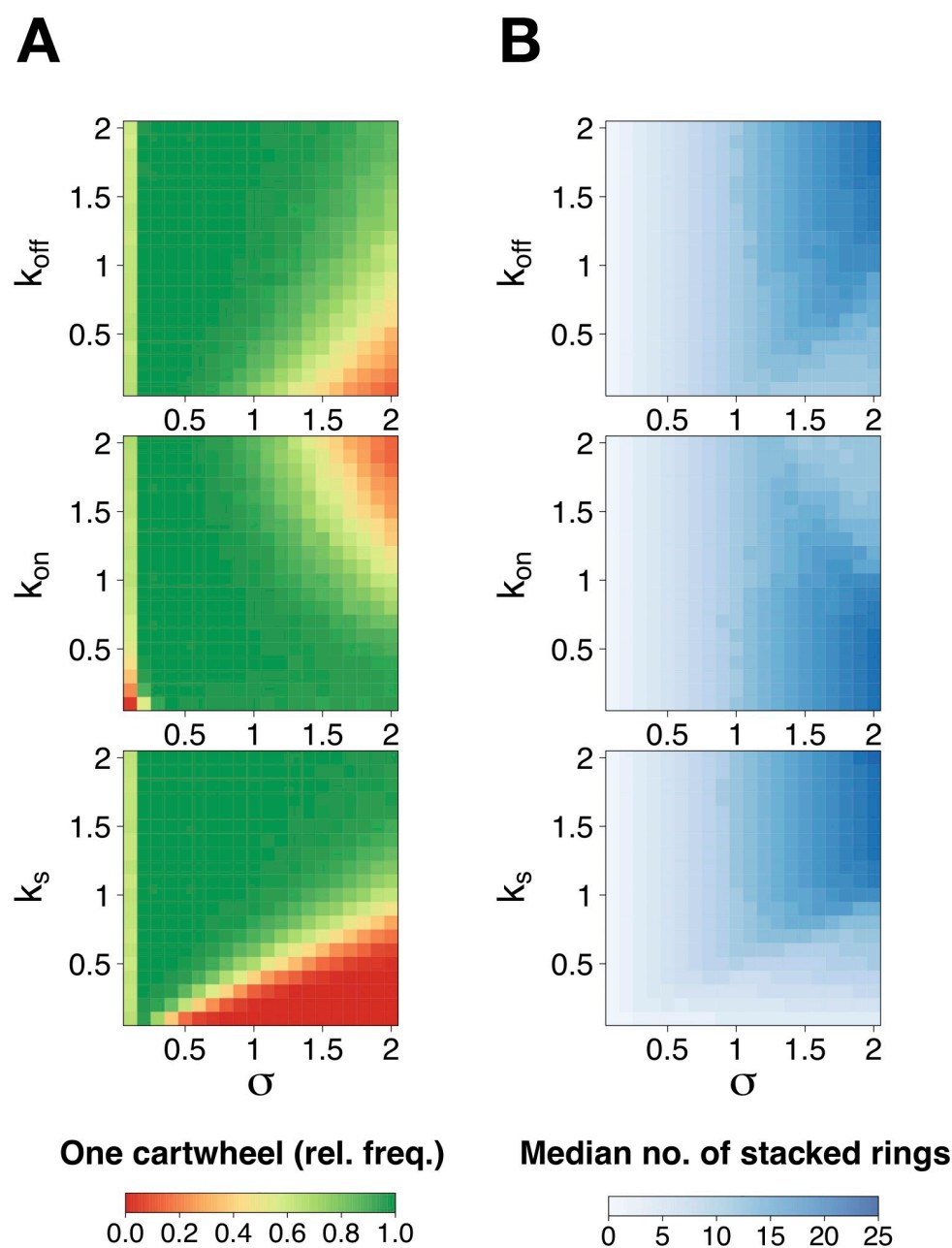

**Fig 5. Probability of forming one and only one cartwheel as a function of influx and reaction parameters.** Relative frequency of simulations containing a single cartwheel at time $t = 100$, under the model assuming irreversible ring assembly and stacking. We varied the indicated parameter values in steps of 0.1, yielding a total of 400 pairwise combinations of parameter values. We used default simulations conditions as indicated in Fig 3 and in the Models and methods section.

reactions. However, it is pertinent to explore a wider region of the parameter space and evaluate model outcomes, to identify the conditions guaranteeing proper cartwheel number control.

We conducted simulations of the model for pairwise combinations of parameter values in [0.1,2.0], whilst keeping other parameters at the value of 1 (Fig 5A and S4 Fig). Specifically, we are interested in characterising the relationship between the influx parameter $\sigma$, that ultimately

limits maximum cartwheel number and length, and the other reaction parameters. In the designated range, there will be, on average, between 10 and 200 SAS-6 dimers at the end of the simulations. This means that 1-22 complete rings may form, on average. It has been estimated by quantitative proteomics that 138 dimers of SAS-6, on average, are present at the centrosome after cartwheel formation, potentially yielding cartwheels of up to 15 complete rings in length [12]. The dimensions of the SAS-6 foci within procentrioles also support an average cartwheel length of 7-17 rings from S through to G2 [37]. Therefore, our analysis covers a range of molecular abundances that can be related to strong SAS-6 depletion through to mild overexpression.

As can be expected, formation of extra cartwheels is favoured for high $k_{on}$/low $k_{off}$ and/or high values of $\sigma$ and inhibited for low $k_{on}$/high $k_{off}$ and/or low $\sigma$. In addition, for low values of and $\sigma$, the formation of even the first cartwheel is almost disabled for concomitantly low $k_{on}$ but seemingly insensitive to changes in $k_{off}$. Nevertheless, cartwheel formation shows greater sensitivity to changes in $k_s$ in comparison to $k_{on}$ and $k_{off}$, for the considered parameter values. As previously observed, the formation of supernumerary cartwheels is promoted for low $k_s$ and hindered for high $k_s$, and decreases sharply as $\sigma$ increases.

With respect to cartwheel elongation, we note once again an opposite trend to cartwheel formation—if fewer cartwheels form, each one elongates at a higher rate (Fig 5B and S5 Fig). Therefore, we obtained higher median cartwheel lengths for high $\sigma$ together with low $k_{on}$, high $k_{off}$ and high $k_s$.

Overall, a single cartwheel is formed in the majority of parameter combinations and for a wide range of SAS-6 levels. In addition, if SAS-6 influx is too low, cartwheel assembly is likely to fail. On the other hand, if it is too high, supernumerary cartwheels become more frequent. These results mimic the effects of SAS-6 depletion or overexpression on centriole duplication, respectively.

Irreversible cartwheel assembly and stacking is a simplifying assumption that facilitates the definition of what constitutes a new and definitive structure, since cartwheels cannot disassemble. While we reasoned that neither process would be biased in this way, it is possible that the assumption of irreversible ring-forming and stacking reactions contributes to robust cartwheel number control. Thus, we modified the model assuming that complete rings can also break, at a rate $k_{off}$, and the top most layer of a stack can dissociate, with a rate $k_u$, (see Models and methods) and asked how the probability of forming one and only one cartwheel varies as a function of model parameters. We set $k_{on} = k_{off} = k_u = 1$ such that ring assembly and disassembly, as well as stacking and "unstacking", occur at equal rates for all combinations of parameter. We tested $k_s = 1$ and $k_s = 5$.

Under identical conditions as before, we observed that the probability of forming one and only one cartwheel is lower when ring formation and stacking are reversible (S6 Fig). Interestingly, our simulations showed that the first cartwheel is more "stable", whereas the second one undergoes frequent assembly disassembly. For example, it appears in 93% of the simulations at some point but is only present at the end of 28%, for $k_s = 1$ (S7A Fig). Formation of one and only one cartwheel can still be achieved by increasing $k_s$ (S7B Fig). Therefore, since these results are not qualitatively different from our previous analysis, we conclude that irreversible ring formation and stacking facilitate but are not strictly necessary for robust cartwheel number control.

Altogether, our analysis reveals that the postulated stacking mechanism allows for cartwheel numbers to be strictly controlled within a broad region of the parameter space, and is robust to stochastic fluctuations in SAS-6 availability. The model behaviour can be summarised as follows: at first, SAS-6 are produced and combine until forming a complete ring which seeds a new cartwheel. Afterwards, the building blocks can be partaken by the existing cartwheel(s) or

give rise to new ones. For the same influx rate, if there are fewer cartwheels, they elongate faster and are likely longer. If there are more cartwheels, they elongate at a slower pace and are likely shorter. Conversely, it is more likely that extra cartwheels are produced if there are fewer cartwheels than if there are more. In other words, the assembly time of additional cartwheels increases with the number of existing cartwheels. In practice, there is an implicit negative feedback on cartwheel assembly as existing cartwheels "consume" intermediates via stacking. Finally, due to stochastic influx of SAS-6 and reaction dynamics, the model predicts variance in both cartwheel numbers and length.

## Comparison between different scenarios for structure number control

Up to now, we have quantified the efficiency of a hypothetical mechanism of number control and its consequences. Since it is currently unknown how cartwheel stacks form, we decided to explore alternative scenarios to better understand the limits of our model and how it could be experimentally verified.

First, a key assumption of the stacking model is that cartwheels can elongate indefinitely. However, it is likely that there are biophysical constraints to cartwheel length. So, we considered a variant of the stacking model where the cartwheel can elongate up to a maximum length. We now assume that cartwheels can contain at most $h$ rings. The stacking reaction onto a specific cartwheel stops when it reaches this maximum length. This variant represents a general scenario where there is some extrinsic or intrinsic upper limit to cartwheel length. The two variants will heretofore be referred to as "unlimited" and "limited" stacking models, respectively.

Second, it has been proposed that the number of subcellular structures can be regulated by self-inhibitory signals [4]. We postulate that cartwheel assembly activates a putative molecule that signals the blockage of additional cartwheel assembly, by inducing SAS-6 degradation or removing it from the centrosome. We assume that such feedback pathway is activated in a gamma-distributed time after the first cartwheel formation event. The use of a gamma-distribution with a shape parameter $\alpha$ and rate parameter $\rho$ reflects a linear signalling cascade with $\alpha$ components, whose residence times are exponentially distributed, at the end of which cartwheel assembly is arrested. This feedback model is to be interpreted as a generic model for structure number control and not as a specific molecular pathway at play during centriole biogenesis.

In a nutshell, the feedback model represents an alternative scenario for structure number control, whereas the limited stacking model features a pertinent alternative to one of the main assumptions of the original model. The main purpose here is to compare the behaviour of these different models with the unlimited stacking model. We performed 1000 simulations of the three models using the default settings ($\sigma = k_{on} = k_{off} = k_s$). In addition, we simulated the models without stacking ($k_s = 0$) and feedback ($\alpha = \rho = 0$). For the feedback model, we set $\alpha = \rho = 1$. This parameterisation yields an exponential distribution with mean 1. In effect, it represents a feedback mechanism enacted by a single component (which "senses" that the first cartwheel has been formed, and shuts down SAS-6 influx) and the best case scenario for structure number control, as feedback is likelier to be almost instantaneous. To clearly separate the unlimited and limited stacking models, we set $h = 6$ when simulating the latter, which should allow for the first cartwheel to attain full length before the simulation terminates.

As cartwheels cannot elongate in the absence of stacking, we reasoned the most general readout to compare between the different models is the distribution of cartwheel assembly times (Fig 6). In the absence of both stacking and feedback, as previously observed, the second cartwheel appears quickly after the first one in all simulations. We observed that the assembly

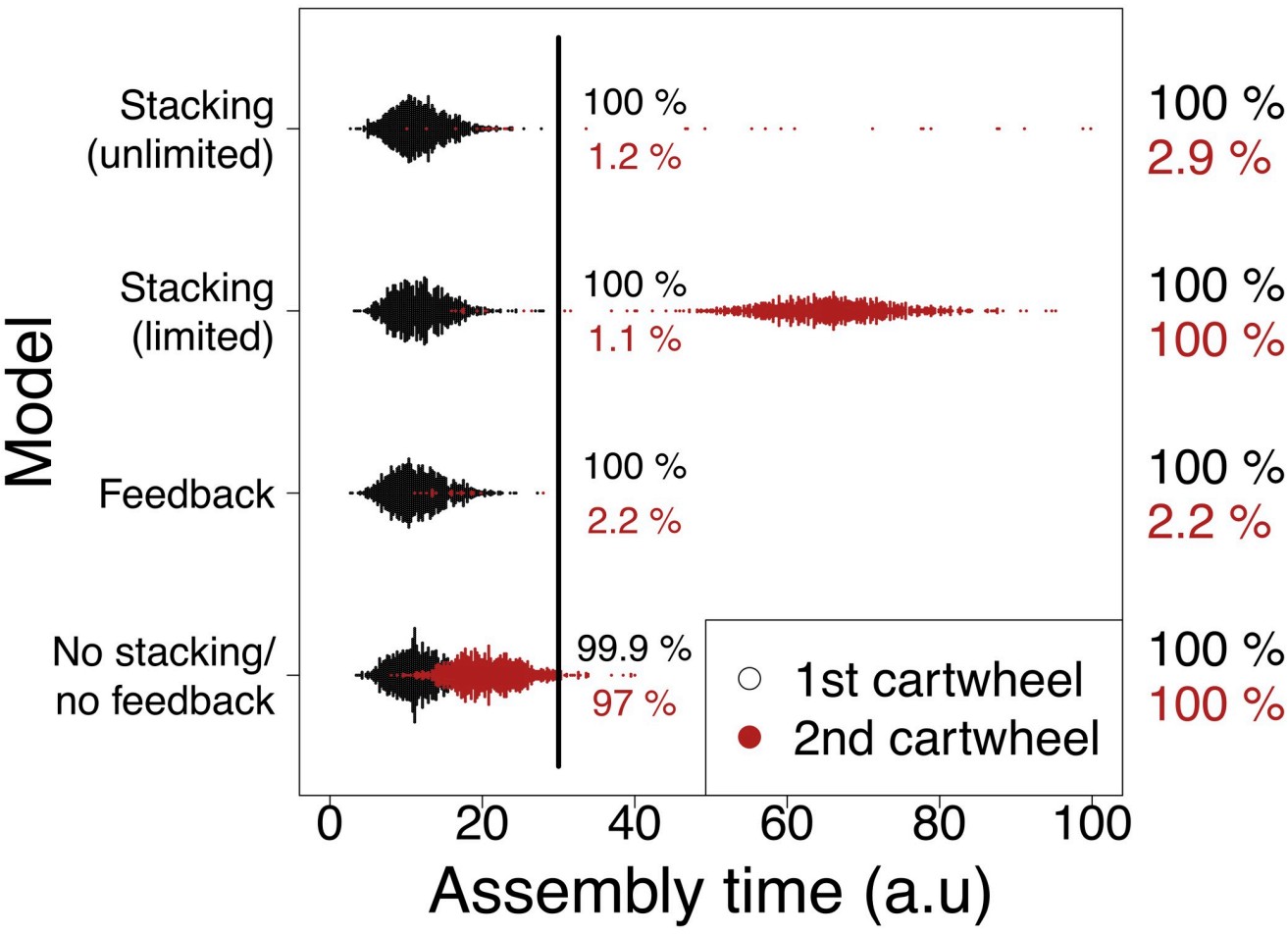

**Fig 6. Assembly times of the first and second cartwheels under different models.** The dots indicate the assembly times of the first (black) and second (red) cartwheels in 1000 simulations of the indicated models. The percentages of first (black) and second (red) cartwheels present at $t = 30$ and when the simulation terminates highlight that supernumerary cartwheels appear earlier in the feedback model when compared to the stacking model(s). We used default simulations conditions as indicated in Fig 3 and in the Models and methods section. For the stacking (limited and unlimited) models $k_s = 1$. Otherwise, $k_s = 0$. In the limited stacking model we set $h = 6$. In the feedback model, we set $\rho = \alpha = 1$.

times of the first and the second cartwheels were similarly distributed. In the limited stacking model, the second cartwheel was formed in 100% of the simulations. As in the unlimited stacking model, there is a time-window during which supernumerary cartwheels scarcely assemble, after which they appear across all simulations following an approximately normal distribution. Indeed, this latter period corresponds to the time when the first cartwheel tends to be fully elongated. By increasing $h$, the distribution of second cartwheel assembly times is shifted towards higher times (S8A Fig). Thus, if $h$ is high enough, the unlimited and limited stacking models will be indistinguishable in a fixed time-window. In this case, cartwheel assembly times are dispersed throughout the whole time-window for cartwheel assembly. In the feedback model, there is an exponentially-distributed time window of opportunity for the appearance of extra cartwheels. Thus, we observed that the second cartwheel is formed relatively early (at $t \leq 30$) and rarely (2.2% of the simulations), marginally outperforming the stacking model. If $\alpha > 1$, mimicking more realistic multi-component linear feedback pathways, the stacking model is more efficient in averting supernumerary cartwheels (S8B and S8C Fig).

In conclusion, the limited stacking model allows for cartwheel number control if $h$ is not reached within the time-window of constant SAS-6 influx. Under these conditions, it is indistinguishable from the unlimited stacking model. Otherwise, supernumerary cartwheels are likely to appear. Note that in this case, cartwheel length may still show variation if one or more stacks of sub-maximal length are built within the permissive time window. Concerning the feedback model, extra cartwheels can only be assembled rapidly after the first one. This marks a difference from the stacking models, under which we expect the assembly time of supernumerary cartwheels to be delayed with respect to the first one.

## Discussion

In this article, we hypothesised that centriole number control can be exerted at the level of cartwheel assembly. We postulated that there is competition between forming new cartwheels and elongating existing ones for a common pool of SAS-6. Using stochastic models, which we solve analytically and with simulations, we showed that if the first cartwheel formed is able to elongate and stack up excess SAS-6 intermediates, this can efficiently prevent formation of extra cartwheels and enable faithful centriole duplication. We showed that the efficiency of this first-takes-all model depends on two fundamental aspects: (1) the length of the sequence of intermediates at which cartwheel assembly can be interrupted by stacking, and (2) the relationship between the SAS-6 influx rate and the stacking rate. For typical nine-fold symmetrical cartwheels, our results indicate that number control can be guaranteed for a wide range of model parameter values. Specifically, the range of $\sigma$ values that produces a single cartwheel copy in simulations indicates that the control mechanism may be robust and insensitive to broadly distributed concentrations per cell of the regulators of the SAS-6 influx. Furthermore, the model makes the non-trivial prediction that cartwheel assembly times should increase with cartwheel numbers and that cartwheel length should be distributed in a cell population, reflecting the variance in SAS-6 influx. These two properties distinguish our model from other commonly cited number control mechanisms.

### Model predictions and limitations

Our model predicts that successive cartwheel formation events are increasingly delayed. In "wild-type" conditions, this means that the first cartwheel is formed quickly but the assembly time of the second one is so long that it falls outside of the time-window that is permissive to centriole biogenesis. In "overexpression" conditions, as in Fig 3, the first few cartwheels form rapidly in succession, new assembly events become rarer and sparser in time. In parallel, existing cartwheels elongate at some constant rate until the next one is formed. Since the model is inherently stochastic, both cartwheel numbers and length are distributed and their probability distribution changes in time. Specifically, both the mean and the variance in cartwheel numbers increase with SAS-6 influx.

The dependence of cartwheel formation on the influx parameter in our model is in agreement with empirical observations of centriole under- and overproduction upon SAS-6 depletion and overexpression, respectively. However, the influx parameter represents in a single average quantity a complex process that includes recruitment via Plk4 and STIL, which is spatially regulated [19, 22]. This implies that the influx parameter does not correspond to SAS-6 expression level directly; rather, it is best approximated by the balance between the rate of SAS-6 loading onto the centrosome, and subsequent turnover, both of which are more challenging to quantify. Furthermore, the levels of Plk4, STIL, and SAS-6 proteins are regulated by degradation, which in some cases can lead to antagonistic interactions, e.g. Plk4 can recruit SAS-6 but may also target it for degradation through FBXW5 [16, 18]. In addition, since the

kinetics of SAS-6 recruitment by Plk4 and STIL are poorly understood, it is possible that they are nonlinear. Altogether, there may be complex interactions between Plk4, STIL, and SAS-6, leading to behaviours that cannot be obtained by assuming that cartwheel number is directly proportional to the total abundance of these proteins in the cell.

Concerning SAS-6 oligomerisation kinetics, the N-terminal interaction of SAS-6 dimers has been reported to be relatively weak [38], so much that biophysical simulations show that complete ring formation is thermodynamically inefficient [36]. Our model is not necessarily inconsistent with these results. Several other factors may contribute to increased stability of SAS-6 complexes *in vivo*, including other partner proteins, such as Cep135/Bld10p [30], the presence of scaffolds [36], molecular crowding [39, 40], or cartwheel stacking itself. In addition, the aforementioned kinetic measurements were based on truncated SAS-6 N-termini, whose behaviour may differ from those of the full protein. Regardless, if none of these factors are limiting for cartwheel formation with respect to SAS-6, then the approximations in our model may still hold.

One of the key aspects of our model is that cartwheel length may be variable under non-strict limits (i.e. there is no maximum length or it is not usually reached at physiological conditions during centriole biogenesis). If, on the other hand, cartwheels reach some maximum length within the time-window where centriole biogenesis is possible, then the stacking mechanism is unlikely to prevent centriole overproduction. Although it is well accepted that average cartwheel length is rather conserved within but may differ between distinct cell types and organisms [41], there is direct and indirect evidence that cartwheel length may be variable. Recent studies found that cartwheel length can vary [30, 42], although it is unclear if the length differences are due to variation in the number of stacked rings. Furthermore, it has been reported that cartwheels undergo length fluctuations during the cell cycle in *Chlamydomonas* and *Spermatozopsis* [43, 44], and disappear during mitosis in human cells (reviewed in [29, 31]). These observations indicate that cartwheel length can vary dynamically. Finally, the length of the SAS-6 focus near centrioles using 3D-STORM has been measured to be within 60-150 nmdepending on the cell cycle stage, suggesting that the cartwheel progressively elongates [37]. Ultimately, cartwheel length regulation is poorly understood but it is central to our hypothetical number control mechanism.

Interestingly, several studies have proposed that SAS-6 homodimers in adjacent cartwheel rings interact via their coiled-coils, forming ring doublets and, possibly, even higher-order assemblies in some species [30, 41, 42]. Thus, it could be that the "minimal cartwheel" is not a single ring, but a ring doublet or a multiple thereof. This scenario can be readily included in our models by assuming that the formation of a ring doublet rather than a single ring, for instance, is irreversible. In these conditions, our hypothetical stacking mechanism should be even more robust in prevent assembly of extra cartwheels, because forming a new one would require eighteen or more SAS-6 homo dimers to join, instead of nine.

Testing our model will require correlating SAS-6 levels with cartwheel numbers and length. Ideally, this should be done at a single-cell level in a synchronised population in G2, i.e. after centriole duplication, following the induction Plk4/STIL/SAS-6 overexpression at different levels. At a finer scale, it requires a readout of cartwheel assembly and stacking kinetics. This is challenging because human cartwheels are extremely small structures. Recent studies succeeded in measuring cartwheel assembly and stacking *in vitro*. First, cartwheel-like stacks were produced and their length measured *in vitro* [30, 32]. Second, a specialised atomic-force microscopy protocol has enabled cartwheel ring assembly to be imaged live [33, 35]. In principle, it should be possible to manipulate the concentration of SAS-6 and study how it affects the number of stacks as well as their length. These systems may be used to study basic properties of cartwheel assembly, as modelled here, bearing in mind that they do not reflect *in vivo*

centriole duplication. Experimental validation of the model lies beyond the scope of this paper but should be addressed in the future. Interestingly, a recent study developed an inducible SAS-6 degradation system with reduced baseline levels of centrosomal SAS-6, which could suggest smaller cartwheels. However, the authors found no numeric centriole assembly defects in these conditions [32]. It could be that between the level of SAS-6 measured in this system and wild-type-like conditions the number of cartwheels is conserved but their length changes, as predicted by our model.

Although the basic properties of SAS-6 self-oligomerisation have been well established in the literature it is still unclear how rings stack up. It has been suggested that rings are first assembled and then loaded onto a growing cartwheel. Such stacking mechanism is distinct from the one we assumed and understanding its consequences for centriole number control would demand specific modelling.

## Generality and relation to other models

A model of cartwheel assembly similar to ours has been recently employed to estimate reaction rates associated SAS-6 self-assembly and dissociation [35]. This model has not been interpreted in the context of centriole number control, let alone the possibility that this is enforced by the unique stacking mechanism we propose. Other models have focused on the issue of copy number control, postulating mechanisms that require a spatially explicit components. In contrast, our model relies only on the existence of multiple intermediates in the cartwheel assembly pathway, which can be sequestered by the elongation of the preformed cartwheels. The sequestering of the building block intermediates represents the inhibitory mechanism that proceeds without demanding the physical existence of a diffusive inhibitor like in lateral inhibition. Since mutants of this diffusive inhibitor will disrupt copy number control in the models of auto-activation and lateral inhibition, we expect our model to be more robust.

Instead, our model is akin to kinetic proofreading. Kinetic proofreading was originally proposed to explain the low error rates observed in DNA replication and protein translation [45], stipulating that errors (i.e. wrong reaction products) would be minimised by a correction mechanism that iterates multiple times over a given substrate. These ideas were later co-opted to explain the narrow specificity of T-cell receptors [46, 47]. In contrast with the latter views, in which TCR signalling is made specific because a multi-iterative process can only be completed in a narrow range of conditions, we take advantage of the complementary wide range of conditions in which such a process fails, preventing extra cartwheels from arising.

As mentioned above, Plk4 and STIL cooperate to shuttle SAS-6 onto the centrosome, and eventually organise into a single focus. Under these conditions, if SAS-6 molecules are locally concentrated, this may favour their aggregation and, importantly, long stacks should be able to elongate more easily and inhibit the formation of supernumerary cartwheels. Therefore, it is likely that Plk4, STIL, and SAS-6 focusing facilitates the stacking process. Note, however, that spatial accumulation of these molecules only ensures that centriole biogenesis should happen preferentially in the focus where they are at supercritical concentrations and not elsewhere. It does not ensure *per se* that one and only one centriole will form at that site without evoking additional control mechanisms, such as stacking mechanism we proposed, or additional spatial constraints such as predefined discrete seeding sites as in [24]. Yet it is pertinent to mention spatial effects that we so far neglected. Diffusion rates of macromolecules are expected to decrease with their oligomerisation state, i.e. larger molecules should diffuse more slowly. Therefore, cartwheels should become increasingly less mobile as they elongate, which should diminish their ability to capture freely-diffusing oligomers and relatively higher stacking rates may be necessary to achieve precise number control. The same holds true if there is any

physical barrier to SAS-6 diffusion (such as the mother centriole itself). Altogether, understanding how this model behaves in spatially explicit conditions is an important question to be addressed in the future. We expect that the mechanism we propose should allow centriole assembly control to be even more robust when Plk4, STIL, and SAS-6 focusing is included explicitly.

Although we discussed our models in the context of duplication of a single pre-existing mother centriole, our results establish how cartwheel number and length depend on SAS-6 availability. One of the predictions of our models is that both the mean and the variance in cartwheel/centriole numbers per cell should increase with higher SAS-6 influx. This is independent of the mode of centriole biogenesis, duplication or *de novo*. Indeed, it has been observed that both the mean and the variance increase in systems where extra centrioles are produced, such as olfactory neuron progenitors [48], where overproduction occurs via rosettes, and other multicilliated cells, where overproduction occurs at deuterosomes or *de novo* [49–52]. In all these cases, the extra centrioles are associated with Plk4 and STIL overexpression, which could lead to an increase in centrosomal SAS-6 influx. Therefore, our putative stacking mechanism is in good agreement with these statistics of centriole numbers. Conversely, our model would fail to explain an increase in the mean centriole number without an increase in variance but, to our knowledge, this has never been observed.

Ultimately, how centrioles duplicate exactly once per cell cycle remains an unresolved question in cell biology. Our model predicts that formation of one and only one cartwheel should be robust to random fluctuations in SAS-6 expression within a cell population. A unique prediction of our model is that such stochasticity should be reflected in cartwheel length variation, which is a property that has been hitherto overlooked, and that both cartwheel numbers and length should increase with SAS-6 influx. Testing these predictions can help further our understanding of how centriole numbers are regulated in proliferating cells. Here, we propose a simple and original mechanism for copy number control that is reminiscent of basic biochemical principles and establishes quantitative relationships between stoichiometry, number, and length.

## Models and methods

In this section we present detailed formulation of the models used in this study and describe the simulation methods.

### SAS-6 dimer fate model

We assume that in the vicinity of a pre-existing mother centriole there is a large enough volume in which the concentrations are isotropic and reaction kinetics can be approximated by mass action, i.e. reaction rates are directly proportional to the concentration of the intervening molecules. We refer to this as the focal volume. This simplification allows us to obtain straightforward analytical solutions and justifies the stochastic simulations described in the next section. FCS data suggests that homodimers are the most abundant SAS-6 species in the cytosol [37]. Thus, we consider in our models that SAS-6 homodimers are the fundamental cartwheel building block. We consider a single SAS-6 homodimer in the presence of a previously formed cartwheel that can elongate. We assume that this dimer can progress sequentially through $i$ intermediate stages, until forming a complete ring of size $r$, thus establishing a new cartwheel. Alternatively, intermediates of size $i$ can be diverted by stacking up on top of the pre-formed

cartwheel. This system can be represented by the following chemical equations:

$$C_i \xrightarrow{k_{on}} C_{i+1}$$

$$C_i \xrightarrow{k_s}$$

(2)

where $C_i$ denotes an intermediate containing $i$ dimers, with $i < r$. We assume that the addition of other dimers to the growing intermediate occurs at a constant rate $k_{on}$, and that stacking of the intermediate on a pre-existing cartwheel occurs at a rate $k_s$. The process stops when the final product $C_r$ is assembled.

Let $T_i$ denote the residence time of the intermediate containing $i$ dimers. We assume all $T_i$ variables are independent and identically distributed, following an exponential distribution with rate parameter $(k_{on} + k_s)$. We define $O_i$ as the sum of residence times $T_1, \ldots, T_i$. The variable $O_i$ is correctly interpreted as the time it takes for completing $i$ reactions. The the cumulative distribution of $O_i$ is given by the convolution of the cumulative distributions of $T_1, \ldots, T_i$. Thus, it can be defined recursively as:

$$F_{O_1}(t) = F_{T_1}(t)$$

$$F_{O_2}(t) = \int_0^t F_{T_2}(t - \tau) F_{O_1}(\tau) \; d\tau$$

$$\ldots$$

$$F_{O_i}(t) = \int_0^t F_{T_i}(t - \tau) F_{O_{i-1}}(\tau) \; d\tau$$

(3)

It is worth noting that these expressions imply that the initial dimer completed at least $i - 1$ reaction steps leading to larger intermediates. In other words, it requires that the initial dimer was not diverted to the elongating chartwheel up to time $t$. Thus, it is convenient to weight Eq 3 by the expected value of the probability that the dimer integrated an $i - 1$ intermediate up to time $t$. This yields:

$$Pr(C_i, t) = \frac{k_{on}^{i-1}}{(k_{on} + k_s)^i} F_{O_{i-1}}(t) - F_{O_i}(t)$$

(4)

$$Pr(C_r, t) = \left( \frac{k_{on}}{k_{on} + k_s} \right)^{r-1} F_{O_{r-1}}(t)$$

(5)

where equation Eq 4 represents the probability distribution that the dimer is in step $i$ of the assembly pathway, at time $t$, and Eq 5 is the probability distribution that the dimer seeded a new cartwheel at time $t$.

Finally, taking the limit of Eq 5 when $t \to \infty$ we obtain:

$$\lim_{t \to \infty} Pr(C_r, t) = \left( \frac{k_{on}}{k_{on} + k_s} \right)^{r-1}$$

(6)

which is the maximal probability that the initial dimer gives rise to an extra cartwheel. This is minimal condition that allows for supernumerary centrioles to be formed.

## Stochastic cartwheel assembly model

We assume a constant influx rate $\sigma$ of SAS-6 dimers, as a simplification of all cellular processes related to SAS-6 expression, degradation, and recruitment. SAS-6 dimers are known to form higher-order assemblies via each of their two N-termini. We describe oligomerisation as a second-order process involving any two intermediates containing $i$ and $j$ dimers, subject to $i + j \leq r$. In other words, oligomers can combine to produce any structure that is not larger than a complete ring. A complete ring is an oligomer with $r$ dimers. It should be noted that ring formation requires $r$ dimers to be bound and an additional reaction to close the ring. For simplicity, we assume this reaction is fast enough that it can be neglected. In addition, we assume that $r$ dimers are irreversibly bound. Thus, a ring effectively gives rise to a new cartwheel. Furthermore, we assume that all oligomerisation reactions, including the ones yielding complete rings, occur at a rate $k_{on}$. Note that the actual oligomerisation reactions include the two exposed N-termini in a given SAS-6 oligomer. Intermediates containing $k < r$ dimers are assumed to dissociate into any of their constituents, containing $i$ and $j$ dimers ($k = i + j$), with a rate $k_{off}$.

In practice, we envision stacking as identical to ring assembly atop a pre-existing cartwheel. This requires vertical interactions between the oligomer and the stack, as well as lateral interactions with other molecules in the top layer of the cartwheel. For simplicity, we assume that this is achieved in a single reaction step, with a rate $k_s$. We assume free-diffusing oligomers containing $i$ dimers can bind to the $j$ dimers in the top layer of the cartwheel, at a rate $k_s$. If $i + j = r$, a new ring is completed on the stack. As is the case with ring formation, stacking reactions are assumed to be irreversible. Technically, our model makes no distinction on the directionality of stacking; we refer to "top" merely as a verbal device for ease of comprehension. Nevertheless, it should be recalled that in some cellular systems cartwheel elongation seems to be proceed unidirectionally from the bottom [53].

This system can be represented by the following chemical equations:

$$\xrightarrow{\sigma} C_1$$

$$C_i + C_j \underset{k_{off}}{\overset{k_{on}}{\rightleftharpoons}} C_{i+j}, \qquad i + j < r$$

$$C_i + C_j \xrightarrow{k_{on}} C_0^*, \qquad i + j = r \qquad\qquad (7)$$

$$C_i^* + C_j \xrightarrow{k_s} C_{i+j}^*, \qquad i + j < r$$

$$C_i^* + C_j \xrightarrow{k_s} C_0^*, \qquad i + j = r$$

using the previous notation and $^*$ to specify stacks. The first equation corresponds to SAS-6 influx; the second equation indicates bimolecular oligomerisation and dissociation; the third equation denotes irreversible assembly of complete rings; the fourth equation represents stacking by combination of a cartwheel with an incomplete top layer (an incomplete stack); the fifth equation represents addition of a complete ring to a stack by adding $r$ dimers to its top layer. Note that the species $C_0^*$ replaces $C_r$ and includes all stacks whose top layer is a complete ring. Thus, cartwheels of different length are not formally represented in this system. However, they can still be tracked in the simulations by considering stacks of different lengths as distinct chemical species.

On a final note, we assumed oligomerisation and stacking is purely arithmetic. In other words, we ignore any steric hindrance between interacting SAS-6 structures. This issue cannot be easily addressed in our discrete stochastic framework but we acknowledge it may be particularly relevant for stacking.

## Variants and alternative models

In the previous model we assumed that ring completion and stacking are irreversible. We relaxed these assumptions to assess their impact on cartwheel formation and elongation. Alternatively, we assumed that complete rings can also dissociate at a constant rate $k_{off}$ and that oligomers can dissociate from the top layer of a stack at a rate $k_u$. Thus, ring disassembly and stack dissociation reactions are considered to be the backward reactions of ring completion through oligomerisation and stacking, respectively (third and fifth equations in system Eq 7.

In the limited stacking model, we set a maximum cartwheel length $h$, which can take positive integer values. In the simulations, we assume that $k_s = 0$ for cartwheels of length $h$.

Regarding the feedback model, we assume that there is an arbitrary number of components that are activated sequentially. If the time for activation of the next component is exponentially distributed, then the time to complete the entire feedback sequence is the sum of each activation time, which is gamma-distributed with shape parameter $\alpha$ and rate $\rho$. In our simulations, we set a flag that is checked once the first cartwheel is formed and triggers the feedback loop. In practice, this involves drawing a random number from a gamma distribution with shape parameter $\alpha$ and $\rho$, after which the simulations are stopped.

## Analysis and simulations

We obtained analytical solutions for the probability distributions of Eqs 4 and 5 at time $t$ using built-in functions in Mathematica. The more realistic models were implemented in Python and simulated with the Gillespie algorithm [54, 55]. No SAS-6 species are present initially in any of the simulations. The parameters are listed S1 Table. Parameter values are indicated where appropriate; by default $\sigma = k_{on} = k_{off} = k_s = 1$ and $r = 9$. Unless stated otherwise, we performed 1000 simulations of the indicated model(s), in the time-window $t = [0, 100]$.

## Supporting information

**S1 Table. Parameters across all models.**
(PDF)

**S1 Fig. Final cartwheel number distributions.** (A-C) Relative frequency of simulations with the indicated number of cartwheels, at the stopping time. We performed 1000 simulations with no SAS-6 molecules initially present and stopped them at time $t = 100$; time $t$ is measured in arbitrary units (a.u.). For all simulations, $k_{on} = k_{off} = 1$. Note that $k_s = 0$ represents absence of stacking. See also Models and methods section for default initial and stopping conditions, and parameter values for the simulations.
(PDF)

**S2 Fig. Final cartwheel length distributions.** (A-C) Relative frequency of simulations where the first (black) and second (red) cartwheels contained the indicated number of stacked rings, at the stopping time. We used default simulation settings as indicated in S1 Fig and described in section Models and methods.
(PDF)

**S3 Fig. The probability of forming one and only one cartwheel increases with ring size.** Relative frequency of simulations containing one and only one cartwheel as a function of time, for the indicated values of *r*. We used default simulation settings as indicated in S1 Fig and described in section Models and methods.
(PDF)

**S4 Fig. Final distribution of cartwheel numbers as a function of influx and different reaction rate parameters.** (A-C) We used default simulation settings as indicated in S1 Fig and described in section Models and methods.
(PDF)

**S5 Fig. Final distribution of cartwheel lengths as a function of influx and different reaction rate parameters.** (A-C) We used default simulation settings as indicated in S1 Fig and described in section Models and methods.
(PDF)

**S6 Fig. Probability of forming one and only one cartwheel as a function of influx and reaction parameters, assuming reversible ring formation and stacking.** Relative frequency of simulations containing a single cartwheel at time $t = 100$, under the model assuming reversible ring assembly and stacking. We varied the parameter values indicated in the x- and y-axes in steps of 0.1, yielding a total of 400 pairwise combinations of parameter values per plot. We used default simulation settings as indicated in S1 Fig and described in section Models and methods, and set $k_u = 1$. Note that $k_s = 0$ represents absence of stacking.
(PDF)

**S7 Fig. The first assembled cartwheel is stably maintained whereas the second one is dynamically assembled and disassembled.** Presence of the first (black) and second (red) cartwheels along time, under the model assuming reversible ring assembly and stacking, in a random sub-sample of 100 out of 1000 simulations. We used default simulation settings as indicated in S1 Fig and described in section Models and methods, and set $k_u = 1$. A—$k_s = 1$; B—$k_s = 5$.
(PDF)

**S8 Fig. Additional features of the limited stacking and feedback models.** (A) Assembly of supernumerary cartwheels is progressively delayed as the maximum cartwheel length $h$ increases. (B) The distribution of feedback times simplifies to an exponential distribution for $\alpha = 1$, with mean $\rho$. Otherwise, it is gamma-distributed. (C) Assembly times of the second cartwheel as a function of feedback parameters. We used default simulation settings as indicated in S1 Fig and described in section Models and methods.
(PDF)

## Acknowledgments

We are grateful to all members of the Quantitative Organism Biology and Cell Cycle Regulation labs for insightful discussions and to Carla A. M. Lopes, Catarina Nabais, Catarina Peneda and Sonia Gomes Pereira, in particular, for reading and commenting an earlier version of the manuscript.

## Author Contributions

**Conceptualization:** Marco António Dias Louro, Mónica Bettencourt-Dias, Jorge Carneiro.

**Formal analysis:** Marco António Dias Louro.

**Funding acquisition:** Mónica Bettencourt-Dias.

**Investigation:** Marco António Dias Louro.

**Supervision:** Mónica Bettencourt-Dias, Jorge Carneiro.

**Writing – original draft:** Marco António Dias Louro, Jorge Carneiro.

**Writing – review & editing:** Marco António Dias Louro, Mónica Bettencourt-Dias, Jorge Carneiro.

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
