## [Decision Letter · Decision Letter 0]

20 Oct 2020

Dear Dr. Carneiro,

Thank you very much for submitting your manuscript "A first-takes-all model of centriole copy number control" for consideration at PLOS Computational Biology.

As with all papers reviewed by the journal, your manuscript was reviewed by members of the editorial board and by several independent reviewers. In light of the reviews (below this email), we would like to invite the resubmission of a significantly-revised version that takes into account the reviewers' comments.

We cannot make any decision about publication until we have seen the revised manuscript and your response to the reviewers' comments. Your revised manuscript is also likely to be sent to reviewers for further evaluation.

Sincerely,

Attila Csikász-Nagy

Associate Editor

PLOS Computational Biology

Mark Alber

Deputy Editor

PLOS Computational Biology

Reviewer's Responses to Questions

**Comments to the Authors:**

Reviewer #1: In this manuscript, Dias Louro M.A. and colleagues proposed a new model to explain how centriole copy number is strictly limited to one. By focusing on the cartwheel assembly and stacking, they generated a mathematical model and used computer simulations for a plausible solution to centriole copy number control. The essence would be that the first cartwheel functions as a template and stacks layers of other cartwheels. Due to this, the formed intermediates of SAS-6 oligomers are incorporated into the first cartwheel pile. Due to the lack of intermediate cartwheel building blocks, second cartwheel pile is not formed at other sites. Based on the models with a certain range of parameters, the authors found the conditions in which only one cartwheel pile is formed within a given assembly time. To understand the principles of centriole copy number control, it is important in the field to raise and compare reasonable theories and models, and test them experimentally. However, this reviewer feels that this model could explain only a small part of the mechanisms of centriole copy number control. It may well explain the dynamics of cartwheel assembly after site selection of centriole duplication, but it would not fit the various observations related to centriole copy number control in the past studies. This reviewer hopes that the following comments will help improve their models in some way.

-Major points

1)The authors should consider the fact that overexpression of Plk4 or STIL also induces centriole overduplication, which might contradict their models. Moreover, SAS-6 loading to centrioles and the cartwheel assembly require Plk4 and STIL. In particular, considering that Plk4 is upstream of the SAS-6 loading, the regulation for centriole copy number is more critically exerted on Plk4. How could the authors fit these observations to their models and simulation results? The authors should discuss this issue in the manuscript.

2)In the models, the authors consider a focal volume around a pre-existing that is large enough to allow for mass action dynamics. What is the basis for this premise? Based on the previous observations, it is likely that SAS-6 dimers are formed in the cytoplasm and targeted into Plk4 around the pre-existing centriole. Although it is possible that SAS-6 influx affects Plk4 dynamics, the site for SAS-6 loading around the pre-existing centriole is basically limited by pre-existing Plk4. Even in this scenario, are their models still feasible?

3)This reviewer assumes that, in reality, the cartwheel stacking ends in the early stage of centriole formation and before genuine centriole elongation, because the outer part of cartwheels is structurally decorated with triplet-microtubules and binding proteins. On the other hand, both SAS-6 total expression level and influx to centrioles are increasing towards late S and G2 phase. This case may correspond to “limited stacking model with high SAS-6 influx”. According to their models and simulations, the risk of forming second cartwheels would be quite high. Is there a way to resolve this discrepancy? In Fig. 6, the authors concluded that “the limited stacking model is OK for cartwheel number control if h is not reached within the time-window of constant SAS-6 influx”, but this condition is unlikely in real centriole formation.

4)Related to 3), the stacked cartwheels are inside and settled at the bottom of an elongating centriole, but how could they interfere with formation of second cartwheel assembly at the opposite side of mother centriole (at least ~200 nm away)? There seems to be structural and distance barriers.

5)Although only with a certain range of parameters (e.g. high Ks and low influx, unlimited or limited with enough h), the stacking model actually works. However, it seems to be more vulnerable to guaranteeing one cartwheel assembly compared to the feedback model. This point should be therefore toned down. In a way, the authors nicely verified these models, but this reviewer is wondering how, in reality, the feedback works to inhibit additional cartwheel formation. The authors stated, “not as a specific molecular pathway at play during centriole biogenesis”, but it would be better to discuss how it could work around the pre-existing centriole. The mechanism by which the cartwheel stacking inhibits the assembly of intermediates at other places is not difficult to imagine, but it is not the case for the feedback model.

-Minor points

6)This reviewer would suggest to change the current title “A first-takes-all model of centriole copy number” which sounds too vague. A first-takes-all is applicable to most of reasonable models and theories to explain the principle of centriole duplication, so the authors should specify the title based on their assumptions and models in this study.

7)The authors should cite "Takao et al., J Cell Biol. 2019", because this is an important paper that conducted mathematical modeling and simulations of centriole duplication.

8)Mislabeled in Fig. 3D.

Reviewer #2: In this work, Marco Louro, Monica Bettencourt-Dias and Jorge Carneiro propose a new theoretical model of centriole copy number control. Centrioles are large assemblies crucial for centrosome and cilia formation. Therefore, the number of centrioles per cell must be tighly controlled, but up to now, this mechanism is not yet clearly understood.

In this interesting new model, the authors propose that the stacking of the cartwheel, which is an assembly made of SAS-6 proteins and at the base of centriole formation, would restrict the number of new centrioles to one copy. It's an extremely interesting model that makes sense with a lot of in vivo observations. Unfortunately, I don't have the skills to properly evaluate the simulations. However I have some centriole related questions :

- First of all, I think this article should be more accessible to biologists. I had a hard time reading the article. I guess it's probably not possible to simplify the text but I think that explanatory scheme of the results could help biologists to understand the results/conclusions.

- In this model, the stacking of the first cartwheel cause depletion of intermediates, blocking the formation of extra cartwheels. This result works very well with canonical duplication but less well with the formation of centrioles in the case of deuterosomes or blepharoplasts, where dozens of cartwheels are formed. In this case, one can imagine that the concentration of SAS-6 is higher and that it would correspond to the "over-expression" condition tested by the author. In fig. 6, in the limited stacking condition, a 2nd cartwheel forms after the first one. What would be the condition for a dozen cartwheels to form? Is it possible and would they form sequentially? In vivo, at the level of the deuterosomes (or the daughter centriole in the same cells), the assembly seems to be almost synchronous.

-Line 370 “Our models predict that formation of one and only one cartwheel should be robust to random fluctuations in SAS-6 expression within a cell population”. It is a very interesting point. Can the authors develop more ? At what concentration could the system no longer control the number of cartwheels? 10 times more, 100, 1000?

-The cartwheel is not only made of rings but also of spokes, probably made of coiled coil domains from different SAS-6 rings. Could this type of organization influence this model? Similarly, the cartwheel is not linear but made of repeating layers. Can the authors comment on this point?

-The authors show that stacking causes depletion of intermediates and inhibits the formation of supernumerary cartwheels. Could this model therefore be applied to the de novo centriole formation ? In the case where there is zero centriole/cartwheel in the cell, a large number of centrioles/cartwheel assemble in the cell and not at the restricted level of a centrosome. Can this model be applied here?

- What do fluctuations in the length of cartwheels mean for the authors? Does this mean that the final length is variable or does the length fluctuate during stacking, such as the dynamic instability of microtubules?

Minor points

-Fig3, the letter of panel “C” should be replace by “D”

-Line 323: There is an extra “the” in this sentence: “some fraction of the this SAS-6”

**Have all data underlying the figures and results presented in the manuscript been provided?**

Reviewer #1: Yes

Reviewer #2: Yes

PLOS authors have the option to publish the peer review history of their article (what does this mean?). If published, this will include your full peer review and any attached files.

Reviewer #1: No

Reviewer #2: No
---

## [Decision Letter · Decision Letter 1]

27 Feb 2021

Dear Dr. Carneiro,

Thank you very much for submitting your manuscript "A first-takes-all model of centriole copy number control based on cartwheel elongation" for consideration at PLOS Computational Biology. As with all papers reviewed by the journal, your manuscript was reviewed by members of the editorial board and by several independent reviewers. The reviewers appreciated the attention to an important topic. Based on the reviews, we are likely to accept this manuscript for publication, providing that you modify the manuscript according to the review recommendations.

Sincerely,

Attila Csikász-Nagy

Associate Editor

PLOS Computational Biology

Mark Alber

Deputy Editor

PLOS Computational Biology

[LINK]

Reviewer's Responses to Questions

**Comments to the Authors:**

Reviewer #1: This reviewer would agree that the authors significantly modified the manuscript following the comments from reviewers. Although no major change was made in the model and simulation data, the premise of the model was totally altered as described below (now in the introduction).

This limitation notwithstanding, the focused accumulation explains naturally the prevention of ectopic centrioles by the maintenance of the key components concentrations below critical values elsewhere in the cell and around the mother centriole. However, these (qualitative or quantitative) proposals fail to explain how the formation of supernumerary centrioles is avoided at the foci, where Plk4, STIL and the other components accumulate at supercritical concentrations. This is the control problem that is solved by the first-takes-all model proposed here.

That is, if this reviewer’s interpretation is correct, it is now assumed that the cartwheel competition occurs only at the focus limited by Plk4/STIL and not at other sites around the mother centriole. The authors also assumed here that Plk4/STIL and other components accumulate there at supercritical concentrations. These important premises should be clearly stated in the abstract. Otherwise, it would be impossible to understand what kind of assumptions this model is based on and what is novel in the model, compared with others. Considering all these things, the current abstract is vague and misleading.

Also, what kind of situation would the authors specifically refer to as supercritical concentration?

In the current abstract, the authors claim ‘we show that this mechanism may ensure formation of one and only one cartwheel over a wide range of parameter values at physiologically relevant conditions.’ Although ‘over a wide range of parameter values’ is tested by simulations, ‘at physiologically relevant conditions’ seems to be overstatement. This is because, as mentioned above, the model itself is assumed under limited conditions.

Reviewer #2: The revision of the manuscript of Dias Louro et al. correctly addresses the questions I raised. Therefore, I recommend this article for publication.

**Have all data underlying the figures and results presented in the manuscript been provided?**

Reviewer #1: Yes

Reviewer #2: Yes

PLOS authors have the option to publish the peer review history of their article (what does this mean?). If published, this will include your full peer review and any attached files.

Reviewer #1: No

Reviewer #2: No

Figure Files:

Data Requirements:

Reproducibility:

References:

---

## [Editor Report · Decision Letter 2]

6 Apr 2021

Dear Dr. Carneiro,

We are pleased to inform you that your manuscript 'A first-takes-all model of centriole copy number control based on cartwheel elongation' has been provisionally accepted for publication in PLOS Computational Biology.

Best regards,

Attila Csikász-Nagy

Associate Editor

PLOS Computational Biology

Mark Alber

Deputy Editor

PLOS Computational Biology

---

## [Editor Report · Acceptance letter]

5 May 2021

PCOMPBIOL-D-20-01695R2 

A first-takes-all model of centriole copy number control based on cartwheel elongation

Dear Dr Carneiro,

I am pleased to inform you that your manuscript has been formally accepted for publication in PLOS Computational Biology. Your manuscript is now with our production department and you will be notified of the publication date in due course.

With kind regards,

Katalin Szabo
